# Next-Generation Sequencing as a Tool to Detect Vaginal Microbiota Disturbances during Pregnancy

**DOI:** 10.3390/microorganisms8111813

**Published:** 2020-11-18

**Authors:** Agnieszka Sroka-Oleksiak, Tomasz Gosiewski, Wojciech Pabian, Artur Gurgul, Przemysław Kapusta, Agnieszka H. Ludwig-Słomczyńska, Paweł P. Wołkow, Monika Brzychczy-Włoch

**Affiliations:** 1Department of Molecular Medical Microbiology, Chair of Microbiology, Faculty of Medicine, Jagiellonian University Medical College, 31-121 Krakow, Poland; agnieszka.sroka@uj.edu.pl (A.S.-O.); tomasz.gosiewski@uj.edu.pl (T.G.); 2Clinical Department of Gynecological Endocrinology and Gynecology, Jagiellonian University Medical College, Kopernika 23, 31-501 Krakow, Poland; mopabian@cyfronet.pl; 3Center for Experimental and Innovative Medicine, University of Agriculture in Krakow, Rędzina 1c, 30-248 Kraków, Poland; artur.gurgul@urk.edu.pl; 4Center for Medical Genomics OMICRON, Jagiellonian University Medical College, Kopernika 7c, 31-034 Krakow, Poland; przemyslaw.kapusta@uj.edu.pl (P.K.); agnieszka.ludwig@uj.edu.pl (A.H.L.-S.); pawel.wolkow@uj.edu.pl (P.P.W.)

**Keywords:** pregnant women, vaginal microbiota, next-generation sequencing, *Lactobacillus* spp., *Streptococcus agalactiae*, GBS, *Gardnerella* spp.

## Abstract

The physiological microbiota of the vagina is responsible for providing a protective barrier, but Some factors can disturb the balance in its composition. At that time, the amounts of the genus *Lactobacillus* decrease, which may lead to the development of infection and severe complications during pregnancy. The aim of the study was the analysis of the bacterial composition of the vagina in 32 Caucasian women at each trimester of pregnancy using the next-generation sequencing method and primers targeting V3-V4 regions. In the studied group, the dominant species were *Lactobacillus iners, Lactobacillus gasseri,* and *Lactobacillus*
*plantarum*. Statistically significant differences in the quantitative composition between trimesters were observed in relation to *Lactobacillus jensenii,*
*Streptococcus agalactiae*, *Lactobacillus iners*, *Gardnerella* spp. Out of the 32 patients, 20 demonstrated fluctuations within the genus *Lactobacillus,* and 9 of them, at different stages of pregnancy, exhibited the presence of potentially pathogenic microbiota, among others: *Streptococcus agalactiae, Gardnerella* spp., *Atopobium vaginae*, and *Enterococcus faecalis.* The composition of the vaginal microbiota during pregnancy was subject to partial changes over trimesters. Although in one-third of the studied patients, both the qualitative and quantitative composition of microbiota was relatively constant, in the remaining patients, physiological and potentially pathogenic fluctuations were distinguished.

## 1. Introduction

The vaginal microbiota is a sustainable ecosystem which plays a crucial role in the prevention of infectious diseases in non-pregnant women as well as during pregnancy (in both maternal and neonatal health) [1,2]. The quantitative and qualitative composition of the bacterial profile in the vagina in most women is dominated by bacteria of the genus *Lactobacillus* [3,4]. The *Lactobacillus* species are able to produce lactic acid, lowering the vaginal pH, therefore, they create a barrier against pathogen invasion and have been regarded as a hallmark of vaginal health [5,6]. Some factors, e.g., antibiotics, douching practices, hormonal disorders, hygiene of intimate parts, systemic diseases, sexual activity, and smoking can predispose to changes in the bacterial composition of the vagina, including colonization with pathogenic microorganisms [6,7]. As a result, the abundance of beneficial lactobacilli decreases [8] and the diversity of streptococci [9], yeast-like fungi (mainly *Candida*) [6], or anaerobic bacteria (e.g., *Gardnerella* spp., *Prevotella*, *Megasphaera*, *Finegoldia,*) [10] increases, which is a convenient condition for the development of vaginal microbiota disturbances or bacterial vaginosis [2,4].

Although epidemiological data indicate disorders in the vaginal microbiota, for example, the presence of group B streptococci (GBS) found in 10–30% of healthy women [11], they do not usually manifest symptoms of infection. These inconspicuous and common microorganisms are dangerous, especially for a developing fetus, because they may lead to premature delivery, miscarriage [8], or other postpartum complications, e.g., neonatal pneumonia, meningitis, and septicemia [3,10]. Therefore, knowledge about the microbiological state of the vagina in each pregnant woman should be necessary.

Until recently, the studies about the composition of the vaginal ecosystem were based on the use of a microbiological culture and phenotype identification of specific species. Actually, these methods can be considered fragmentary because they provide only limited information on bacterial communities [4]. Moreover, many aerobic and anaerobic bacteria are difficult to culture, or there is a lack of appropriate phenotypic tests to identify them. More accurate identification can be obtained using high throughput analyses, e.g., next-generation sequencing (NGS). This type of research uses partial primer sequences that attach to the highly conserved hypervariable regions of the bacterial 16S rRNA gene [1]. The obtained amplicons are sequenced and then subjected to phylogenetic analysis consisting of assigning a sequence to a specific unit at the selected taxonomic level (e.g., family, genus, or species). Studies on the human microbiome with the application of this method have definitely shown more richness in the microbiota (both aerobic and anaerobic) than those identified using culturing methods [12]. For this reason, the implementation of the NGS seems to be an appropriate method to monitor the changes in the vaginal composition of bacteria during pregnancy and its relationship with susceptibility to infection, the possibility of premature birth, and postpartum complications [3,6,13,14].

The aim of our study was an evaluation of the semi-quantitative and qualitative dynamics of vaginal microbiota of healthy Caucasian women at the beginning of each trimester of pregnancy with the use of next-generation sequencing.

## 2. Materials and Methods

### 2.1. Patients

The study included 32 healthy pregnant women aged 22–35 years old, without clinical genitourinary symptoms of infection. Detailed inclusion and exclusion criteria are shown in Table 1. The consent obtained from the participants was both informed and written. The study has been approved by the Bioethics Committee (No. KBET/47/B/2009—samples collection and No. KBET/1072.6120.51.2017—standardization of molecular diagnostic methods), and the patients tested gave their written consent for participation in the study. Two swabs from the lower vagina (vaginal introitus), were taken from patients during a routine prenatal visit in each trimester of pregnancy. The remaining information about the patients has been described previously in an article in which the samples were used for research based on culture methods [15].

### 2.2. Samples

The vaginal swabs were placed in a non-nutrient Amies transport medium (Eurotubo) and delivered to the Department of Microbiology, Jagiellonian University Medical College, within about 2 h. Each time, the first vaginal swab was used to prepare a smear stained with the Gram method, to evaluate the vaginal flora condition according to the 10-point Nugent score [16]. The second swab was suspended in 1 mL of 0.9% NaCl, vortexed for 1 min, and stored at −70 °C. From the material obtained, the volume of 500 µL was used in the previous study for quantitative and qualitative assessment of culture [15]. The remaining 500 µL was used for the present research. Bacterial DNA was isolated from the samples using the procedure based on enzymatic and mechanic lysis, described by Gosiewski [17], following the manufacturer’s protocol for the Mini genomic DNA isolation kit (A&A Biotechnology, Gdynia, Poland). Additionally, along with 20 µL lysozyme and 10 µL lysostaphin, 10 µL of mutanolysin was added to the samples during the enzymatic lysis step. The purity and concentration of the isolates obtained were measured with the use of a NanoDrop spectrophotometer (Thermo Scientific, Waltham, MA, USA).

### 2.3. Library Preparation

The DNA extracted was used to carry out PCR amplification (T100 Thermal Cycler, BioRad), with primers targeting the 16S rRNA gene V3 and V4 regions, primer F: TCGTCGGCAGCGTCAGATGTGTATAAGAGACAGCCTACGGGNGGCWGCAG, and primer R: GTCTCGTGGGCTCGGAGATGTGTATAAGAGACAGGACTACHVGGGTATCTAATCC (adapter sequences are highlighted in gray). The sequences of primers, the composition of the reaction mixture, and the program for amplification are shown in Table 2.

The volume of 5 µL of each amplicon was subjected to electrophoretic separation on 1.5% agarose gel (Prona ABO, Gdańsk, Poland) diluted 10 × by TBE buffer (Sigma-Aldrich). The PCR products (size, ~550 bp) were visualized in the Quantity One (BioRad, Hercules, CA, USA) application in the presence of UV light. Subsequent steps (purifying, samples indexing, samples quantification, and pooling) were prepared with the use of 16S Metagenomic sequencing library preparation protocol (Illumina, San Diego, CA, USA) [18].

### 2.4. Next-Generation Sequencing

The pooled library, with a 10% spike-in PhiX control DNA was applied to the cartridge for sequencing. Sequencing was performed using the Reagent Kit V3 (600 cycles) in the MiSeq platform (Illumina) in the Center for Medical Genomics OMICRON, Jagiellonian University Medical College, Krakow, Poland.

### 2.5. Bioinformatics Analysis

Raw sequencing reads were controlled for quality using FastQC software (Babraham Bioinformatics). No significant loss in read quality, and no overrepresented primers or adapters sequences were detected. The reads were further analyzed using BaseSpace (Illumina San Diego, CA, USA) 16S Metagenomics application, which is a high-performance implementation of the Ribosomal Database Project (RDP) Classifier described by Wang Q. et al. [19]. In brief, the analysis included matching the reads to primer sequences, excluding non-target reads, filtering the low quality reads by base-call quality, length, and ambiguity, and merging the paired-end reads. Then, chimeric reads were detected using the UCHIME tool [20] and further assigned to the taxonomic classes using the RDP algorithm based on the Bayesian approach [19]. The RDP classification was made with respect to the RefSeq RDP 16S v3 database [21]. This version of the database contains 14,676 bacterial and 660 archaea full 16S rRNA gene sequences. Counts of reads classified to the specific taxonomic units were used to assess alpha and beta diversity using Microbiome Analyst software [22]. Alpha diversity, expressed in the Shannon index, was assessed for significance using Kruskal–Wallis test. The beta diversity was evaluated using the Shannon index and was subsequently tested for significance using PRMANOVA, and visualized with PCoA. To further analyze differences in amounts of separate taxa in the study groups, DeSeq2 [23] software was used to normalize read counts and to perform differential analysis. Obtained *p*-values were corrected for multiple testing using the false discovery rate (FDR) procedure [24]. Differences in microbiota profile in separate patients among trimesters were tested for statistical significance using Fisher exact test in contingency tables and two-sided *p*-values. Dependencies between microbiota within separate samples among trimesters were analyzed using the Spearman rank correlation coefficient. Differences in other parameters, such as age, pH, and Nugent among groups, were evaluated using ANOVA analysis. The later statistical analyses were performed using JASP software, version 0.11.1 [25]. The values presented in brackets refer to the arithmetic mean and standard deviation. In every case, *p*-values < 0.05 were considered statistically significant.

## 3. Results

### 3.1. Characteristics of the Study Population

Thirty-two pregnant women were included in the present study. The mean age in the experimental group was 30.2 ± 3.9 years. The remaining features were measured in each trimester: Nugent score (the mean value in pregnant women in trimester I was 1.3 ± 1.4, 1.0 ± 1.3 in trimester II and 1.6 ± 1.8 in trimester III) and pH (the mean was 4.7 ± 0.7, 4.9 ± 0.4 in trimester I and II and 4.8 ± 0.5 in trimester III). There was no statistical significance between the three trimesters concerning the above-described features (*p* = 1.00).

### 3.2. Metagenomic Sequencing

The data obtained by sequencing consisted of 10,123,878 reads (19,869 minimum reads per sample, 221,589 maximum, and 114,116 median) with an average number of reads of 115,044. A phylogenetic summary of the results is presented in Table 3.

Alpha diversity, expressed as the Shannon index and beta diversity (PCoA) in subsequent trimesters, was similar and not statistically significant (respectively: *p* = 0.493 and *p* = 0.845), Figure 1.

### 3.3. Analysis of the Vaginal Microbiota in Each Trimester in the Pregnant Women Group

Assessment of the vaginal microbiota was conducted at six taxonomic levels (phylum, class, order, family, genus, species). Three of them: L2 (phylum), L6 (genus), and L7 (species) are described in this publication. At the phylum level (L2), all the OTUs obtained were assigned to four phyla: *Firmicutes, Actinobacteria, Proteobacteria,* and *Bacteroidetes.* The phylum *Firmicutes* was clearly predominant in each trimester (95.43%), while other phyla constituted negligible amounts (2.81%, 1.27%, and 0.35%, respectively), as shown in Figure 2. The values in parentheses represent the average percentage of bacteria at a specific taxonomic level throughout pregnancy. There were no statistically significant differences between profiles of vaginal microbiota in subsequent trimesters at the phylum level.

At the genus level (L6), the OTUs identified corresponded to 115 genera, but only 5 of them were present in more than 1% and common to all trimesters: *Lactobacillus*, *Streptococcus*; *Bifidobacterium, Gardnerella,* and *Escherichia* (Figure 3).

Other genera accounted for less than 1% (e.g., Prevotella, Staphylococcus, Lactococcus, Bacteroides, Finegoldia, Ureaplasma, Citrobacter, and Faecalibacterium). The abundance of the genus Lactobacillus was >90% in 23 pregnant women.

Statistically significant differences were found between 2nd and 3rd trimesters with regard to the abundance of the genus *Streptococcus* (*p* < 0.01) and also between all trimesters in relation to the genus *Gardnerella* (*p* < 0.01).

At level L7, 2535 species were identified, but each trimester in the studied group was dominated by species of the genus *Lactobacillus*: *L. iners* (52.26%) *L. gasseri* (21.28%)*, L. plantarum* (9.14%), *L. acidophilus* (3.04%), and *L. jensenii* (3.00%). Other species with a percentage greater than 1% were: *Gardnerella* spp. (1.99%), *Streptococcus agalactiae* (1.31%), and *Bifidobacterium breve* (1.17%) (Figure 4).

The remaining species were present in negligible amounts. Statistically significant differences were observed between the 1st and 2nd trimesters with regard to the amounts of *S. agalactiae* and *L. iners* (*p* < 0.001), and between the 1st and 3rd with regard to the amounts of *Gardnerella* spp. (*p* < 0.001). There were also statistically significant differences in the abundance of *L. jensenii* over the three trimesters (*p* < 0.01).

### 3.4. Semi-Quantitative and Qualitative Composition of the Physiological Microbiota in Individual Patients

In the next part of the study, the vaginal microbiota from the three trimesters was analyzed individually for each patient. Based on the data obtained, the samples were divided into three categories:

#### 3.4.1. Stable Microbiota during Pregnancy

Out of the 32 women, 12 (37.5%) had a relatively stable composition of the vaginal microbiota during the whole period of pregnancy, and there were no statistically significant differences between species diversity and trimesters. Among these patients, 6 (18.75%) were characterized by the domination of *L. iners*, which accounted for 60% to 99% of all species. In the remaining patients (43.75%), other species of the genus *Lactobacillus* were in a similar percentage: *L. gasseri* (20–35%), *L. acidophilus* (13–26%), and *L. helveticus* (8–21%).

#### 3.4.2. Fluctuations in the Non-Pathogenic Potential *Lactobacillus* Species Composition of the Vaginal Microbiota

Bacterial profiles of 9 (28.1%) more women showed the presence of species fluctuations during the pregnancy, mainly within the genus *Lactobacillus*. For example, in one patient, the number of *L. jensenii* in the 1st trimester was 95.7%, while in subsequent trimesters, this species was replaced mainly by *L. gasseri* (87.1% in 2nd trimester and 97.3% in 3rd trimester). In five patients (15.63%), a high negative, but not significant, Spearman’s correlation (−0.976 to −0.997) was observed between the amount of *L. iners* and *L. gasseri*. In three patients, an increase was observed in the percentage of *L. iners* over 3 trimesters (average percentage in the 1st trimester, 15.57%; 82.1% in the 2nd and 75.05% in the 3rd) and a simultaneous decrease in the percentage of *L. gasseri* (71.54% in the 1st, 6.26% in the 2nd and 1.25% in the 3rd). The opposite relationship was observed in the other two patients: a decrease in the *L. iners* percentage (69.7% in the 1st, 12.75% in the 2nd and 0.11% in the 3rd) was accompanied by an increase in *L. gasseri* (7.85% in the 1st, 20.15% in the 2nd, 23.71% in the the 3rd trimester). However, probably due to the small number of observations, the determined correlations were statistically insignificant, but it is worth paying attention to in the context of future research. On the other hand, in the three remaining patients, similar relationships to those described above were noted in relation to *L. iners* and *L. plantarum*.

#### 3.4.3. Fluctuations in the Pathogenic Potential Species Composition of the Vaginal Microbiota

In 11 women, a significant presence of vaginal microbiota with pathogenic potential was observed across the trimesters. The fluctuations were mainly related to a significant increase in *Gardnerella* spp., *S. agalactiae*, *E. faecalis*, and *A. vaginae*. Due to the large amount of data, detailed observations of bacterial profiles of only several patients are shown and described in Figure 5.

During the first trimester of pregnancy, the vagina of patient number 18 was colonized mainly by *L. plantarum* and other species of the genus *Lactobacillus*. In the second trimester, this trend shifted to *L. iners, L. acidophilus,* and *L. helveticus,* and also, a small amount of *Gardnerella* spp. appeared. In the last trimester, the percentage of *L. plantarum* was restored, but significant amounts of *Gardnerella* spp. were also observed (*p* < 0.01). In patient number 38, both in the first as well as the second trimesters, high percentages of *S. agalactiae* were noted (*p* < 0.01). In the third trimester, the bacterial balance was restored and the percentage of *S. agalactiae* was reduced to trace amounts. The next example of bacterial dynamics in the vagina is patient number 10. In the first trimester, her vaginal microbiota was homogeneous, dominated by *L. gasseri*. In the second trimester, *B. breve* and *S. anginosus* additionally appeared, but in the third trimester, *p. bivia* and *S. anginosus* reached statistically significant amounts compared to earlier trimesters (*p* < 0.01). Moreover, the amounts of *L. gasseri* were reduced almost five times (*p* < 0.01). Although in patient number 7, it is difficult to reach conclusions about the dynamics, the bacterial profile was distinctly different from other patients. In all three trimesters, a considerable and significant percentage (*p* = 0.03) of *Gardnerella* spp. was present.

Patient number 6: During the 1st trimester, the vagina of the patient was colonised mainly by *Enterococcus faecalis* and *Staphylococcus saprophyticus* (*p* < 0.01). In the 2nd and 3rd trimester these bacteria were observed in trace amounts. In relation to *E. faecalis*, a similar result was observed in another patient with pathogenic potential fluctuations.

Patient number 9: in the 1st trimester, low amounts of *Gardnerella* spp. and high amounts of GBS were observed, but in the 2nd and 3rd trimesters a reverse observation (also significant, *p* < 0.03) was noted. In relation to GBS, similar results were observed in 2 other patients. 

Patient number 16: in the 1st trimester, low amounts of *Gardenrella* spp. and *Atopobium vaginae* were noted, but these values in the 2nd and 3rd trimesters (both in relations to *Gardnerella* spp. and only in the 3rd in relations to *Atopobium vaginae*) were significantly higher (*p* < 0.01). 

Patient number 42: a high percent of *Lactobacillus gasseri* was observed in the 1st and 2nd trimesters, but in 3rd, a significant decrease in the amounts of *L. gasseri* species. In the 3rd trimester, considerable and statistically significant percent of GBS appeared (*p* < 0.001). 

## 4. Discussion

In our study, we demonstrated the semi-quantitative and qualitative composition of the vaginal microbiota during pregnancy in healthy Caucasian women, with the use of the next-generation sequencing method. The main limitation of the study was the small study cohort, therefore, the presented results require confirmation in the future in a larger research group. The strength of our research was the analysis of the bacterial composition at three-time points for one ethnic group and also for each patient individually. Until now, only a few works in the available literature have obtained samples from two or more trimesters and used the sequencing method [26,27,28,29,30]. In the study by Romero et al., although samples were taken, on average, six times during the whole pregnancy, the results were mainly related to African Americans (*n* = 19). Other ethnic groups that were isolated were White (*n* = 2) and Hispanic (*n* = 1) [26]. It was similar in another work by the same authors, where out of 72 women who delivered at term, only four were White. The vaginal microbiota of 12 pregnant women at four-time points (8–12, 17–21, 27–31, and 36–38 weeks of gestation) were also subject to the research of the team headed by Walther-Antonio et al. [30]. In the work of these authors, the attention was paid to differences in the bacterial vaginal composition between Caucasian and African-American pregnant women in relation to the findings of Romero et al. [30].

Our study results are in accordance with previously published works, in which it is emphasized that, during pregnancy, the physiological vaginal microbiota is dominated by the genus *Lactobacillus* [10,26,27,28,29,31]. The dominance of this genus in gestating women is associated with increased production of estrogen. As a consequence, glycogen is accumulated and metabolized to lactic acid, which promotes the proliferation of lactobacilli [9,26,32].

It is worth noting that among the genus *Lactobacillus,* not all species are equal. The deficiency of *L. iners* and or *L. gasseri* may lead to greater susceptibility to pathogenic changes of the microbiota. However, this is not a rule. In our work, in relation to the analyzed group, *L. iners* was predominant among this genus, while others (*L. acidophilus, L. helveticus*, and *L. gasseri*) occurred in smaller amounts. In a systematic review, van de Wijgert et al. observed that, in the majority of studies, one cluster was dominant, i.e., *L. crispatus* (11 articles) or *L. iners* (15 articles), other species are less common [3]—data from 2014. Some authors have observed that *L. iners* prevails more often in Black African and African American women and also is connected with greater bacterial diversity compared to Caucasian or Asian women. In the latter group, *L. crispatus, L. jensenii, L. gasseri,* and/or *L. vaginalis* are predominant [26]. Walther-Antonio et al. suggest that the dominance of *L. crispatus* and *L. iners* is dependent on the pregnant woman’s age, and *L. iners* is dominant in older gravidae (34–36 years old). Unfortunately, these observations are based on two units only [30]. The role of these species in the vagina during pregnancy is still unclear, but several articles suggest that *L. iners* can be dominant at the time when the microbiota is in the transition phase between “normal” and “abnormal” [33] or helps to restore the domination of Lactobacilli after antibiotic treatment [3]. This is also likely due to the fact that the *L. iners* genome also encodes inerolysin, a pore-forming toxin (related to *Gardnerella* spp. vaginolysin), and has clonal variants that promote vaginal health in some cases and are associated with dysbiosis and disease in others [34].

Another work suggests that *L. iners* is the only species of the genus *Lactobacillus*, which after using antibiotics, can dominate the bacterial microbiota of the vagina. Consequently, it can make patients more susceptible to new episodes of bacterial vaginal infections [35]. Moreover, it is associated with a higher prevalence of sexually transmitted infections [36].

In our research, only a small percentage of *L. crispatus* (from 0.53 to 0.58%) was identified in relation to the above-mentioned ones. These results are surprising, especially when combined with previously published analyses carried out on the same experimental group but using Sanger sequencing [37]. In the previous study, the dominant species was *L. crispatus* (29% in the 1st trimester, 51.6% in the 2nd and 25.8% in the 3rd trimester); the second most frequent species was *L. gasseri* (19.4%, 25.8%, and 25.8%, respectively) and then *L. rhamnosus, L. amylovorus,* and *L. johnsonii. L. iners* was not identified. Significant discrepancies between the results of the studies were obtained, which mainly stemmed from preparing the material from patients differently. In previous studies, the isolation of bacterial DNA was carried out on the basis of colonies grown only on solid MRS medium (Difco), and in this study, DNA was isolated from swabs suspended in physiological saline. The species *L. iners* is difficult to culture on MRS medium and requires the components contained in the blood medium for growth. As a result, it was not possible to observe the growth of this species and its identification. These studies show the limitations of culture and the advantages of molecular methods, but the latter are not without their drawbacks, which is discussed later on in the discussion.

The conducted studies showed that, during pregnancy, there are changes in the qualitative and semi-quantitative composition of bacteria in the vagina over the course of 3 trimesters at the genus and species level. Although in 12 out of 32 patients, these changes were not statistically significant, in the remaining patients, significant fluctuations in numbers and qualities, physiological or pathogenic in nature, were found. An interesting observation was to show a strong but insignificant (probably due to the small number of observations) negative correlation between *L. iners* and *L. gasseri* in the group of patients with physiological fluctuations. So far, no similar dependencies have been found in the available literature. These two species are present both in healthy pregnant women and those with dysbiosis [35] and are claimed to have a low impact on the stability of the microbiota. Moreover, in the work of Verstraelen et al., *L. iners* and *L. gasseri* are also mentioned as species that, with time, may strongly predispose to bacterial overgrowth in the vagina during pregnancy [35]. The above statements may justify the following successive results of our studies regarding patients in whom qualitative and/or semi-quantitative vaginal microbiota with pathogenic potential fluctuations have been observed during different periods of pregnancy. The samples in which the dominant species of the genus *Lactobacillus* was *L. gasseri* were characterized by a much greater diversity and percentage share of other bacteria, e.g., *Gardnerella* spp., *S. agalactiae, Streptococcus anginosus,* and *Prevotella bivia.*

Unfortunately, the exact role of *L. gasseri* is still unknown. Interesting results were published in the work by Atassi et al., where it was proved on the basis of a series of in-vitro experiments that some *L. gasseri* strains at pH > 4 show a bactericidal effect against *S. agalactiae* [38]. These conclusions may explain the pathogenic fluctuations observed in this publication in patient no 38 (Figure 5). Although a significant percentage of GBS (31% and 58%, respectively) and a relatively low *L. gasseri* (14% and 16%) were recorded in the first and second trimesters, these values reversed in the third trimester in favor of the species *L. gasseri* (86%), unrivaled in the domination of the vaginal microbiota.

In the remaining patients with pathogenic fluctuations, in whom significant percentages of *Lactobacillus* species other than *L. gasseri* were found, typical bacteria described in the course of bacterial vaginosis (BV) were also identified: *Gardnerella* spp. and *Atopobium vaginale*. It should be noted here that none of the examined patients showed symptoms typical of BV. Although in the case of *Gardnerella* spp., the literature data indicate that, in as many as 80% of women, colonization with this bacterium is asymptomatic [38]. Moreover, on the basis of the research of Wong et al., the high incidence of *Gardnerella* spp. in women without confirmed BV does not refute the hypothesis that *Gardnerella* spp. is the causative factor for BV. The functional role of *Gardnerella* spp. in the vagina can vary greatly depending on the strains of these bacteria in health or illness [39,40]. On the other hand, the increased abundance of *Gardnerella* spp. leads to the formation of a proteolytic enzyme-rich biofilm on the surface of epithelial cells, which causes epithelial exfoliation, creating colonization sites for other anaerobes or facultative anaerobes, e.g., *Atopobium vaginae*. Further changes in the vaginal ecosystem characteristic for BV may predispose to infections with other microorganisms, e.g., *Trichomonas vaginalis*, *Neisseria gonorrhoeae,* and *Chlamydia trachomatis* [41].

In some cases, in our study, the abundance of bacteria of the genus *Bifidobacterium (B. breve, B. longum* and *B. bifidum*) was also found, especially in the 3rd trimester. It is probably an effect of progesterone, the principal gestational hormone, which reaches its highest concentrations in the last period of pregnancy and promotes an increase in the relative abundance of *Bifidobacterium*. Research by the team of Nuriel-Ohayon et al. in pregnant women, and also in a mouse model and under in vitro conditions, confirms these findings [42]. Species of the genus *Bifidobacterium* play an important role, not only in the pregnant mother (boosting the immune system, improving insulin sensitivity) but first of all in the infant. They are passed on to the newborn during natural birth and during breastfeeding and are the major type in the healthy infant’s digestive tract [43]. Bifidobacteria play an important role in the maturation of the immune system early in life and produce lactic acid, which has the ability to metabolize human milk oligosaccharides. Reduced numbers of these bacteria in infants are associated with disease states [44]. Therefore, it can be assumed that the increase in the number of bacteria of the genus *Bifidobacterium* in the third trimester of pregnancy is associated with natural preparation for childbirth.

While *Bifidobacterium* is considered to be a component of the natural bacterial microbiota, *Streptococcus* and *Prevotella* are often found in bacterial disorders or bacterial vaginosis (BV) [45]. However, their presence should not always be associated with a pathogenic condition or infection (as above for *Gardnerella* spp.), because not every deviation from the normal composition of the vaginal bacterial microbiota indicates disease. For its development, the interaction between virulence factors and their quantitative domination is necessary. Accordingly, it can be argued that some taxa, depending on certain factors, may act as commensals or pathogens [46]. Moreover, some bacteria, which are also present in BV, e.g., *Megasphaera* or *Atopobium*, have the ability to produce lactic acid, which would indicate that it is possible for species other than lactobacilli to protect one against the proliferation of pathogenic bacteria [32]. For this reason, it is necessary to be able to properly interpret these data.

In the next-generation sequencing research, apart from the fact that interpretation of the results is often complicated, the methodology is also an important issue. First of all, the selection of suitable primer pairs and hypervariable regions of the bacterial 16S rRNA gene is significant and may have influence on obtaining different results. In our study, universal primers from the Illumina protocol were used, covering regions V3–V4, similarly to the research of Borgdorff et al. [36]. In the literature, we found information stating that for both short and longer read sequences, region V4, which is also included in our set of primers, is appropriate for capturing microbial diversity [5]. The results of our research showed that, with the use of these primer pairs, it is possible to reliably identify all groups of bacteria which are reported in similar studies. However, it would be necessary to perform comparative studies in the future using primers amplifying other hypervariable regions of the bacterial 16S rRNA gene. The problem of choosing the proper primer pairs in research on the vaginal microbiota is quite a widely discussed issue. Aagard, DiGiulio and Walther-Antonio and co-workers, in their research, used primers covering regions V3–V5 [27,29,30], Subramaniam et al.: V4 [5], Fettweis and Virtanen with co-workers: V1–V3 [6,47], MacIntyre and Romero and co-workers: V1–V2 [10,26]. Although each of these regions will allow insight into the bacterial community in the sample, it should be noted that the results will be flawed. For instance, the set of primers V1–V3 makes it possible to distinguish between *Lactobacillus* species but may underestimate the genera *Acinetobacter* and *Escherichia.* Moreover, it does not fully distinguish the family *Enterobacteriaceae* and some other genera, for example, *Staphylococcus*. Other sets of primers, V3–V5, are better for the study of *Enterobacteriaceae* and *Bifidobacteriaceae* but not suited to identify the species from *Lactobacillus* and *Prevotella* genus [9]. Similarly, the set of primers V1–V2 allows to identify some phylotypes at the genus level, but fewer at the species level [26]. Until now, no optimal solution has been found, so further research is needed to select the right, universal primers to study the vaginal microbiota.

Another important methodological issue is the selection of tools and databases for comparing and assigning sequences to the appropriate taxonomic levels with the increasing emphasis on species-level classification. The most commonly used tools are: RDP Classifier [48], 16S Classifier [49], Kraken classifier, GOTTCHA, LMAT, OneCodex [50] SPINGO, SILVA [51], and Greengenes [52], software with the classification option, e.g., QIIME [53] and MG-RAST [54]. Despite the large selection and general availability of these tools, classification at the species level still presents problems, which are widely discussed, among others, in the works of Gao et al. [54] and Tuzhikov et al. [55]. A comparative study conducted by Lindgreen et al. showed that none of these classifiers could be used as the best choice for sequence analysis [56]. In our research, there was only 47–57% sequence alignment at the species level, which is a much lower percentage compared to the higher taxonomic levels (Table 3).

Next-generation sequencing requires further improvement and unification of some tools that allow proper determination of the vaginal taxonomic composition, primarily the development of universal primers and the creation of one reference database. Despite these limitations, NGS is currently the quickest and most accurate solution in vaginal microbiota testing compared to culture methods. Moreover, it can also be useful in difficult cases when multiple bacterial agents are involved.

## 5. Conclusions

In our study, we have shown that the vaginal microbiota in the group of pregnant women is subject to partial changes during the three trimesters of pregnancy. Although these changes were not reflected in the values of the alpha and beta diversity parameters, they were significant, especially at the genus and species level. In the group of patients with physiological fluctuations, they mainly concerned *Lactobacillus iners*, and in patients with pathogenic fluctuations, *Streptococcus agalactiae* and *Gardnerella* spp. The most abundant genus *Lactobacillus* was represented mainly by *Lactobacillus iners*. Moreover, an interesting trend was observed, which concerned a strong negative but insignificant correlation between the abundance of *L. iners* and *L. gasseri.* It is worth paying attention to this relationship in the context of future research.

## Figures and Tables

**Figure 1 microorganisms-08-01813-f001:**
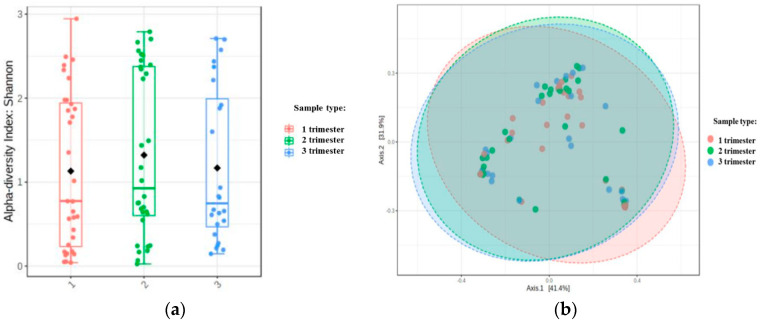
Alpha and beta diversity expressed as Shannon index (**a**) and PCoA Jensen-Shannon index (**b**).

**Figure 2 microorganisms-08-01813-f002:**
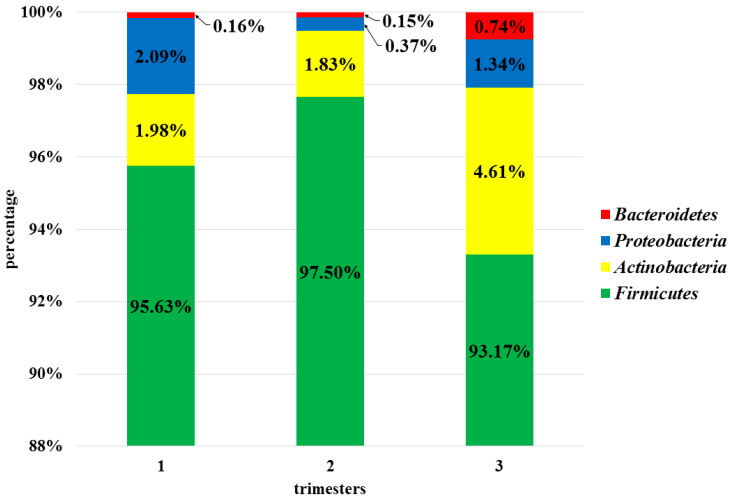
Bacterial profiles for 1st, 2nd, and 3rd trimesters at the phylum level (L2).

**Figure 3 microorganisms-08-01813-f003:**
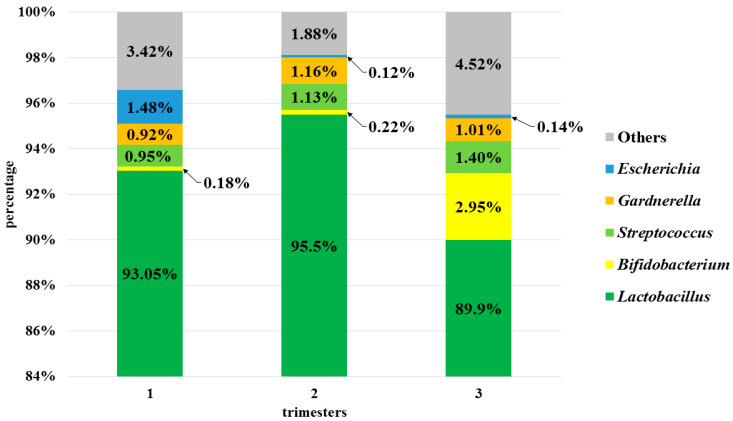
Bacterial profiles for 1st, 2nd and 3rd trimesters at the genus level (L6).

**Figure 4 microorganisms-08-01813-f004:**
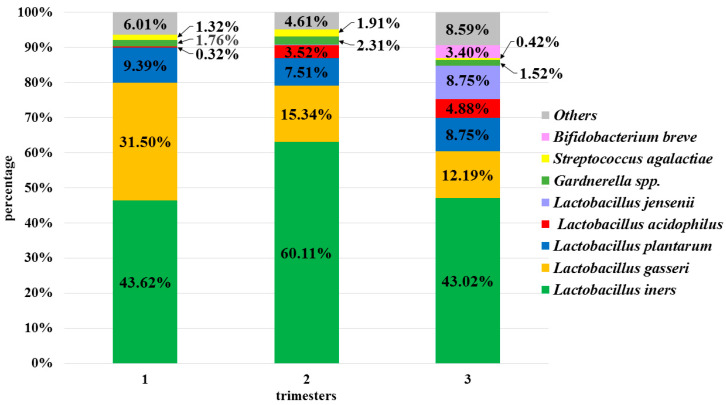
Bacterial profiles for 1st, 2nd and 3rd trimesters at the species level (L7).

**Figure 5 microorganisms-08-01813-f005:**
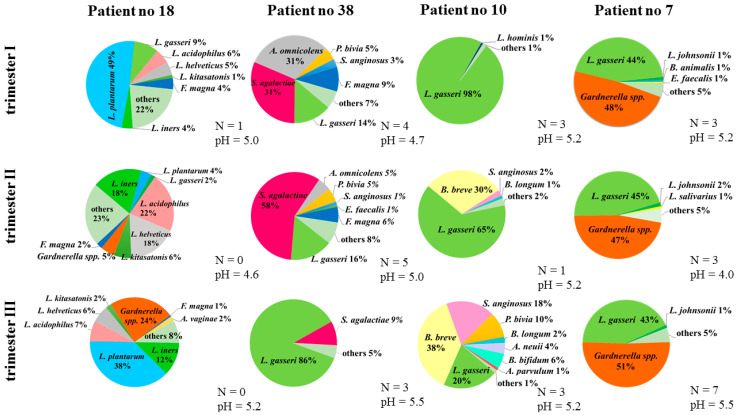
Bacterial profiles of selected patients, whose samples were clearly different from other pregnant women. For each sample and trimester, values of pH and Nugent score (N) are provided under the pie chart.

**Table 1 microorganisms-08-01813-t001:** Inclusion and exclusion criteria used for the recruitment of patients into the study.

Inclusion Criteria	Exclusion Criteria
- women in the first trimester of pregnancy, aged 18–40 years - absence of clinical signs of urogenital infection - lack of antibiotic or probiotic use for up to 30 days before getting pregnant and during pregnancy - value of 0–6 at the 10 -point Nugent score in the first trimester as a confirmation of physiological flora of the genitourinary tract- written consent to participate in the study	- pregnant women under the age of 18 and over 40 years old - women with a high-risk pregnancy - rupture of the membranes - gestational diabetes - use of antibiotics for up to 30 days before becoming pregnant or during pregnancy - diagnosis of bacterial vaginosis - result of 7–10 at the 10-point Nugent score in the first trimester - clinical symptoms of urinary tract infection - lack of written consent to participate in the research

**Table 2 microorganisms-08-01813-t002:** The composition of the reaction mixtures, the reagents involved, and PCR reaction thermal profiles.

Final Volume: 25 µL	Thermal Profile
H_2_O	10.5 µL	95 °C -	5 min
Kapa Biosystems (Roche)	12.5 µL	95 °C -	30 s	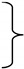	
Primer 1 (F) (Genomed)	0.5 µL	55 °C -	30 s	30 ×
Primer 2 (R) (Genomed)	0.5 µL	72 °C -	30 s	
DNA	1.0 µL	72 °C -	5 min

**Table 3 microorganisms-08-01813-t003:** A phylogenetic summary of the results obtained.

Taxonomic Level	Percent ^1^ of Reads in 1st Trimester	Percent ^1^ of Reads in 2nd Trimester	Percent ^1^ of Reads in 3rd Trimester
kingdom	98.82%	98.17%	98.32%
phylum	98.72%	98.07%	98.19%
class	98.66%	98.01%	98.11%
order	98.58%	97.95%	98.02%
family	98.47%	97.85%	97.89%
genus	98.11%	97.55%	97.14%
species	47.58%	45.22%	57.09%

^1^ Percentage of reads assigned to the appropriate systematic levels.

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
