# Peer review of "Next-Generation Sequencing as a Tool to Detect Vaginal Microbiota Disturbances during Pregnancy"

_microorganisms, 2020, doi:10.3390/microorganisms8111813_

Round 1

Reviewer 1 Report

This manuscript by Sroka-Oleksiak et al. describes the semi-quantitative and qualitative analysis of the composition of vaginal microbiota at three points of pregnancy for one ethnic group (Caucasian). Overall, the study shows that the composition of the vaginal microbiota during pregnancy is subject to partial changes over trimesters. The manuscript is well written, with clear and nicely explained data, and is easy to follow. The findings are thoroughly discussed in the context of previous studies. I only have a few issues that require addressing:

  1. While lactobacillus is known to decrease the pH, what is the effect of other bacteria that were identified on the vaginal pH? Did the variations in those bacterial populations influence the vaginal pH?
  2. Since variations in the pathogenic bacteria were observed between trimesters, were there any associated clinical observations?
  3. Line 142: replace “sing” with “using”.

Reviewer 2 Report

This manuscript describe the application of next-generation sequencing technique applied to a study cohort of Caucasian pregnant women. The study was performed parallel to a previous published paper from the same research group. The latter consisted in the quantitative and qualitative assessment of vaginal swab using culture based methods focusing mainly on the role of group B streptococci. The extended analysis of this subsequent paper give a broader overview of the vaginal bacterial composition and its fluctuations during pregnancy.

Overall, the manuscript is original and encourage the development of a suitable tool for pregnancy screening and follow-up visits. The paper is well organized and well written whit detailed methods description. As mentioned by authors, the strength of the research is the time-point analysis of vaginal microbiome that could be useful in predicting complications during pregnancy. However, the limitations of the study have to be highlighted since a small study cohort respect to other published papers could be misleading. Furthermore, a correlation whit the pregnancy outcome is necessary since the bacterial composition of some individuals seems to be predictive for some complications. The authors are encouraged in focusing on microbiota fluctuations group.

Introduction

  1. Family, genus and species of bacteria must be written in italics. Please check the entire manuscript (e. g. lines 40, 209, 379)
  2. Line 47: It is well known that the change in the bacterial composition during bacterial vaginosis consists in the overgrowth of several anaerobic bacteria forming a polymicrobial biofilm. Please remove “an” from the sentence.

 Materials and methods

  1. Table 2: Please write “H2O” correctly.

Results

  1. Figure 1: The quality of the figure must be improved. Maybe the poor quality is due to the built-in PDF.

The observed increase of G. vaginalis and A. vaginae in the microbiota fluctuations group may not be unexpected since the prevalence of L. iners is somehow often linked to vaginal dysbiosis. Figure 5 represents only 4 women out of 11, and patient number 7 clearly show a bacterial vaginosis pattern especially in the 3rd trimester. The authors know that the presence of these bacteria are detrimental in pregnancy outcome. Are the other 7 patients not relevant? A brief description could be interesting to clarify relationships between beneficial and potentially pathogenic species. These results are undoubtedly useful and have to be described.

Reviewer 3 Report

Major revision:

It is not clear that the authors suggest being the principal aim of the manuscript: if it is a methodological issue (the title refers to it), it lacks a comparative analysis. If the authors aimed to analyze vaginal microbiota composition (or its perturbations) during pregnancy, that lacks a novelty as there are many publications including about microbiota in a European population (MacIntyre et al., 2015). I suggest that the authors reorganize and /or supplement the manuscript to clarify the principal idea (or ideas) revealing the novelty and value of the research.

Minor revisions:

Gardnerella vaginalis should be replaced with Gardnerella spp. due to the revision in Gardnerella taxonomy (Vaneechoutte et al., 2019).

Ref. 14 is the meeting abstract, but not a paper.

Round 2

Reviewer 3 Report

The authors responded to my comments and included additional information in the Discussion section.